# Adult Langerhans Cell Histiocytosis and the Skeleton

**DOI:** 10.3390/jcm11040909

**Published:** 2022-02-09

**Authors:** Danae Georgakopoulou, Athanasios D. Anastasilakis, Polyzois Makras

**Affiliations:** 1LCH Adult Clinic, 251 Hellenic Air Force & VA General Hospital, 11525 Athens, Greece; danaigeorgakopoulou@hotmail.com; 2Department of Endocrinology, 424 General Military Hospital, 56429 Thessaloniki, Greece; a.anastasilakis@gmail.com; 3Department of Medical Research, 251 Hellenic Air Force & VA General Hospital, 11525 Athens, Greece

**Keywords:** Langerhans cell histiocytosis (LCH), bone, treatment, skeleton

## Abstract

Langerhans cell histiocytosis (LCH) is a rare inflammatory neoplasia in which somatic mutations in components of the MAPK/ERK pathway have been identified. Osseous involvement is evident in approximately 80% of all patients and may present as a single osteolytic lesion, as a multi-ostotic single system disease or as part of multisystem disease. Both exogenous, such as treatment with glucocorticoids, and endogenous parameters, such as anterior pituitary hormone deficiencies and inflammatory cytokines, may severely affect bone metabolism in LCH. Computed tomography (CT) or magnetic resonance imaging (MRI) are usually required to precisely assess the degree of bone involvement; 18F-fluorodeoxyglucose (FDG) positron emission tomography—CT can both detect otherwise undetectable LCH lesions and differentiate metabolically active from inactive or resolved disease, while concomitantly being useful in the assessment of treatment response. Treatment of skeletal involvement may vary depending on location, extent, size, and symptoms of the disease from close observation and follow-up in unifocal single-system disease to chemotherapy and gene-targeted treatment in cases with multisystem involvement. In any case of osseous involvement, bisphosphonates might be considered as a treatment option especially if pain relief is urgently needed. Finally, a patient-specific approach is suggested to avoid unnecessary extensive surgical interventions and/or medical overtreatment.

## 1. Introduction

Langerhans cell histiocytosis (LCH) is an orphan disease characterized by clonal proliferation of abnormal CD1a+/CD207+ myeloid precursors cells. The abnormal cells in LCH are actually derived from myeloid dendritic cells that exhibit the same antigens (CD1a, S100 protein, and Langerin) and the same unique intracytoplasmic organelles (Birbeck granules) with the normal Langerhans cells found in the skin and mucosa [1].

The fundamental nature of LCH as a neoplastic versus reactive disorder has been an ongoing debate and its pathogenesis still remains unclear. The recent identification of somatic mutations in components of the MAPK/ERK pathway point towards a neoplastic rather than a reactive process and the World Health Organization has recently recognized LCH as a hematopoietic neoplasm. The LCH cells are characterized by activation of the MAPK/ERK pathway secondary to a cancer-associated mutation (BRAFV600E) in approximately 60% of LCH lesions [1,2]. In addition to this BRAF mutation, a high prevalence of somatic MAP2K1 mutations has been reported in BRAF V600E–negative Langerhans cell histiocytosis [3]. Recent studies suggest BRAF V600E mutation as a significant predictor of disease progression and as a candidate for targeted therapy in patients with disease relapse or multisystem disease [4].

The inflammatory component defines some of the most significant acute and long-term manifestations of the disease. A combining hypothesis may consider LCH as an inflammatory neoplasia with pathological LCH cells proliferating and accumulating in LCH lesions because of oncogenic mutations while concomitantly recruiting and activating inflammatory cells, including T-lymphocytes, macrophages, plasma cells, eosinophils, neutrophils, natural killer cells and osteoclast-like multinucleated giant cells (MGCs) [5]. Confirmation of LCH requires proof of a clonal neoplastic proliferation with positive staining for CD1a, S100 and langerin (CD207); the expression of CD68 is variable and CD163 stain is usually absent [6].

Although it is more commonly diagnosed in childhood, LCH may emerge at any age and with various degrees of systemic involvement [7]. The Histiocyte Society (HS) classifies the clinical forms of LCH according to the number and type of organs involved: single system (SS) LCH if one organ/system is involved (either unifocal or multifocal) and multisystem (MS) LCH if two or more organs/systems are involved (with or without risk-organ involvement). Bones, skin, lungs, and pituitary gland are the most commonly involved sites while lesions at lymph nodes, liver, spleen, intestine and central nervous system (CNS) are less frequent. Clinical manifestations vary according to the tissue involved, and the course of the disease ranges from a self-limiting condition to a chronic disease with remissions and recurrences [8]. Involvement of risk organs (bone marrow, CNS, liver or spleen) is associated with a worse prognosis [8].

This review aims to describe the pathogenesis, presentation, and evaluation of osseous involvement among adults with LCH. Moreover, various therapeutic strategies for bone lesions are described according to the status and extent of the disease.

## 2. Bone Metabolism in LCH

There are several factors that may affect the clinical course of LCH such as treatment with glucocorticoids, anterior pituitary hormone deficiencies, and the inflammatory cytokines which provoke lesional and systemic alterations of various components known to negatively affect bone metabolism (Figure 1).

Systemic glucocorticoids are commonly used in the nodular form of pulmonary LCH for symptomatic patients in whom smoking cessation fails to improve lung function or respiratory symptoms. The combination of vinblastine/prednisone is the standard treatment for children with LCH and remains an option for adult MS LCH, SS-LCH with multifocal lesions, and SS-LCH with “special site” lesions [8]. However, glucocorticoids have a well-known adverse effect on growth in children and on bone metabolism, bone mass, and strength in both children and adults [9]. Immunosuppressive drugs, such as methotrexate, which are used as “mild systemic therapy”, may also induce bone growth arrest and osteoporosis through the inhibition of osteoblast proliferation and differentiation [10].

LCH shows a great predilection for the hypothalamus–pituitary (HP) axis, resulting in permanent pituitary dysfunction in up to 20% of patients with LCH [11,12]. Pituitary hormone deficiencies, mostly attributed to pituitary and/or stalk infiltration, including central hypogonadism, central hypothyroidism, growth hormone (GH) deficiency and hyperprolactinemia, are either directly or indirectly associated with altered bone metabolism [8,13,14]. Therefore, it seems rational to conclude that an HP-related effect on bone metabolism might be quite common in LCH cases, especially among adults, and mainly through a dysregulated secretion of gonadotropins and a low level of gonadal steroids in both males and females [15].

Besides LCH cells, disease lesions also contain several inflammatory cell populations, including T lymphocytes macrophages, plasma cells, eosinophils, osteoclast-like (OC-like) multinucleate giant cells and neutrophils. These infiltrating cells and LCH cells stimulate each other to produce a milieu of cytokines, such as granulocyte-macrophage colony-stimulating factor (GM-CSF), interferon gamma (IFN-γ), tumor necrosis factors TNFs, interleukin (IL) -1, IL-2, IL-3, IL-4, IL-5, IL-6, 1L-7, 1L-10, IL-17, receptor activator of nuclear factor kappa-Β ligand (RANKL) and osteoprotegerin (OPG) [16]. The proinflammatory cytokines IL-1α, IL-1β, TNF-α, and TNF-β are potent stimulators of bone resorption and inhibitors of bone formation; IL-6 increases osteoclastogenesis in cell cultures and may mediate some of the resorbing activity of PTH whereas inflammatory cytokines such as IL-7 and IL-17 also stimulate bone resorption [17].

In LCH, OC-like multinucleated giant cells are present not only in osseous but also in non-osseous lesions, such as skin and lymph node infiltrations, and can be activated by molecules such as RANKL and macrophage colony-stimulating factor (M-CSF) produced by both LCH cells and T cells. In addition, the lesional and systemic interplay between T cells and dendritic cells through RANK/RANKL interaction may promote the growth and activation of T cells and enhance the activation and survival of dendritic cells [18]. Moreover, it has been suggested that dendritic cells could differentiate into OC-like cells [19]. Cytokines derived from OC-like cells could also play a role in the chronic tissue destruction and the development of osteolytic bone lesions [16,20]. Additionally, pathological LCH cells are probably lacking the protective role in bone tissue homeostasis of normal Langerhans cells [18,21].

A role of RANKL in the pathogenesis of LCH has recently been introduced. Adult patients with LCH have higher serum OPG and lower serum RANKL levels than controls regardless of bone involvement [20] while among children with active LCH a positive correlation was reported between RANKL/OPG ratio and osteolytic activity [22]. In the adult study, these changes have been independently associated with the disease and it has been hypothesized that serum RANKL levels are lower because of a shift of circulating RANKL to LCH lesions with a concomitant increase in its cell-bound concentrations, and a compensatory increase of OPG levels [20]. In support of this hypothesis, RANKL was found to be abundantly expressed in cells within inflammatory infiltrates of various osseous and non-osseous LCH lesions, which in turn may activate OC-like multinucleated giant cells [23], pointing out RANKL inhibition as a rational therapeutic approach for the management of the disease.

In an effort to find a serum marker of LCH activity periostin, a secreted extracellular matrix protein expressed in collagen-rich connective tissues and especially in bone, was investigated. In the skeletal environment, periostin is both a signaling molecule, enhancing osteoblastic activity and bone formation via the inhibition of sclerostin and the consequent activation of the Wnt/β-catenin pathway, and a structural component of bone matrix [24]. Lower serum periostin levels among adult patients with active LCH than controls have been recently reported independently of markers of bone remodeling and skeletal infiltration and it was speculated that downregulation of periostin is a compensatory mechanism to prevent fibrosis in this disease [25]. Therefore, it was suggested that periostin is a potential serum biomarker of LCH activity and needs further investigation as it is also the case in several other diseases like asthma and a number of neoplasms [26]. In the same study, no difference in sclerostin levels was found between LCH patients and controls; moreover, the typical inverse correlation between sclerostin and periostin was not evident in LCH patients, therefore pointing to a bone-independent mechanism of periostin reduction.

Besides the focal bone involvement in form of bone lesions, LCH seems to affect the entire skeleton through alterations of bone metabolism. We have previously shown that 20% of all adult patients with LCH have a bone mineral density (BMD) measurement below the expected age range (Z-score ≤ −2.0). Furthermore, patients with active disease have lower BMD compared both to controls and patients with inactive disease. Treatment of LCH in our study was associated with reduced bone metabolism, as both osteoblastic (P1NP) and osteoclastic (CTX) serum markers levels were significantly lower in patients who had previously received chemotherapy and/or glucocorticoids compared to those who received no systemic treatment [27].

## 3. Skeletal Involvement in LCH

### 3.1. Disease Manifestations

Osseous involvement may present as a single osteolytic lesion, as a multi-ostotic single system disease or as part of multisystem disease. Bone lesions are present in approximately 80% of all patients with LCH (30–50% in adults) and most often manifest with focal pain and abnormal growth of soft tissue adjacent to the affected bone (Figure 2a) [28,29]. Clinical examination usually reveals a soft and sensitive protuberance. The skull and chest wall are the most commonly affected bones in both adults and children, with the spine and long bones in children, and the spine and jaw in adults being the second most affected skeletal sites [30].

Vertebral lesions with intraspinal projection or craniofacial bone lesions with soft tissue extensions are considered “special site lesions” because of the risk of developing spinal cord compression and CNS infiltration, respectively. In particular, patients with craniofacial bone lytic lesions are at increased risk of developing HP axis damage, space-occupying CNS lesions, and neurodegenerative disease [11,18]. Moreover, involvement of the orbital bones may be associated with exophthalmos and involvement of the temporal bone with mastoiditis and chronic otitis [8].

Oral cavity involvement, with or without mandibular infiltration, occurs early in LCH, but it is frequently misdiagnosed as the initial symptoms may be non-specific [31]. Involvement of the alveolar bone leads to a “tooth floating in air” appearance, causing tooth mobility and displacement, thereby mimicking periodontitis [31]. Mucosal lesions are usually localized in the buccal mucosa and at the back of the vestibule. They are painful, ulcerated, ovoid or round lesions with erythematous, inflamed borders. Periodontal lesions such as gingival inflammation, ulceration, destruction of the keratinized gingiva, gingival recession, periodontal pockets and bleeding of the oral soft tissues can be a consequence of the alveolar bone loss [32].

### 3.2. Evaluation

Radiographs during the active phase typically show lytic lesions without marginal sclerosis or periosteal reaction [33]. As lesions become chronic, they may resolve or appear sclerotic due to periosteal new bone formation. Therefore, a chronic lesion will typically have a sharply defined sclerotic margin and this can be regarded as a sign of the healing process [33]. Although in LCH with bone lesions the imaging findings are non-specific, isolated diaphyseal destruction of a long bone with fusiform periosteal reaction and peripheral oedema, vertebra plana of the spine, and the bevelled edge of the skull defects accompanied by soft tissue masses strongly suggest LCH [34]. Bone scintigraphy with technitium 99 m was used quite frequently in the past to detect bone lesions but it is not now considered a sensitive diagnostic approach and it is not anymore recommended in the evaluation of osseous involvement [35]. Computed tomography (CT) (Figure 2b) or magnetic resonance imaging (MRI) (Figure 2a) may be required to assess precisely the degree of trabecular and cortical bone destruction in areas at risk of fracture, and to guide a bone biopsy if necessary (CT), or to assess the degree of soft tissue infiltration in areas at risk of neurological complications (MRI) [36]. In addition, whole-body MRI could prove very useful in the detection of small, otherwise undetectable, vertebral lesions [37]. 18F-fluorodeoxyglucose (FDG) positron emission tomography-computed tomography (FDG-PET/CT) could detect LCH lesions not detectable with other imaging modalities (CT, MRI) and could differentiate metabolically active from inactive or resolved disease, information that would preclude biopsy or treatment in metabolically inactive lesions and is essential for prognosis [38]. Moreover, FDG-PET/CT is very useful in the follow-up of patients with positive findings at the time of diagnosis and can be repeated after approximately three months following the initiation of treatment in order to assess the response.

## 4. Treatment

The changes in the management of LCH with bone involvement over the years reflect the changing concepts regarding the disease pathogenesis. Unlike pediatric LCH treatment, a standard first-line chemotherapy treatment has not been established for adult LCH patients. Glucocorticoids, immunosuppressive agents, chemotherapy, and antiresorptives may be used in different regimens and doses depending on the age, the number and location of the lesions and the symptoms of the patient (Table 1). Radiation or surgery may also be performed on specific occasions.

### 4.1. Unifocal Skeletal Involvement

In case of unifocal bone involvement of “non-CNS-risk facial bones” in SS LCH, treatment may vary depending on location, size, and symptoms of the disease from close observation and follow-up (after biopsy or curettage), to radiation therapy, or intralesional injection of glucocorticoids (e.g., methylprednisolone 40–160 mg, or triamcinolone) [39,40]. Biopsy or curettage can sometimes be enough to initiate a healing process; complete surgical removal is also an option, although it may sometimes increase the healing time and/or leave a large bone deficit that would be difficult to be filled [18]. Radiation therapy (e.g., 2 × 10 Gy) is indicated in cases of an impending neurological deficit among patients of high surgical risk or in cases of an accompanying soft tissue component [41]. Antiresorptive agents, such as bisphosphonates, could also be considered [42]. Patients with lesions of the facial bones or anterior or middle cranial fossae with intracranial tumor extension comprise a CNS-risk group. These patients have an increased risk of developing other CNS diseases and DI so systemic treatment is highly recommended [43].

### 4.2. Single System Polyostotic Disease

For patients with multifocal bone LCH, bisphosphonates can be used as the initial treatment approach [42] and in case of no response radiation is the second step if 1–2 lesions while systemic therapy should be preferred in >2 lesions [43]. Chemotherapy is also frequently used. Cytosine arabinoside (ARA-C) has been proposed as the most effective and least toxic regimen for the treatment of LCH bone lesions [44].

A retrospective international study that included both children and adults concluded that bisphosphonates (alendronate, pamidronate, or zoledronate) are effective in active bone lesions, especially in pain resolution, with less adverse effects than other systemic treatments; zoledronate was the most frequently used regimen in this study and almost half of the cases received a single infusion [42]. The resolution of bone lesions is explained by the anti-osteoclastic effect of bisphosphonates that helps reduce the noxious inflammatory substances and other degrading cytokines in the active lesions [42]. Importantly, a beneficial effect of bisphosphonate treatment in co-existing non-osseous LCH lesions has also been reported [42]; moreover, indomethacin is also suggested as an alternative therapeutic approach for patients with primary and recurrent bone involvement [43].

As detailed above, denosumab, a human antibody against RANKL, could be a rational treatment option in LCH, even in extraosseous disease, in order to contribute and further support the endogenous OPG action and control the lesional immunological process [23,45]. A phase IIb clinical trial testing the effectiveness of denosumab (four doses of 120 mg every eight weeks) in adult LCH patients (NCT03270020) is ongoing.

### 4.3. Multisystem Disease

For patients with bone disease within an MS symptomatic setting, first-line treatment options include chemotherapy with cytarabine, cladribine (2-CdA), methotrexate or methotrexate plus cytarabine, or hydroxyurea given that LCH shares both features of an inflammatory and neoplastic disease [43]. The combination of vinblastine and prednisone therapy has never been prospectively proven as effective among adults, in contrast with pediatric LCH [46,47]. Additionally, in adult cases, steroid therapy has been associated with considerable toxicity and vinblastine with frequent neuropathy [44].

Recent studies suggest 2-CdA monotherapy as an effective treatment option with a high overall response rate of around 80–90% [48,49]. Importantly, in nearly two-thirds of the patients, the responses were maintained for at least five years of follow-up [48,49]. Responses were irrespective of disease sites and BRAF-V600E status [49] although a trend toward better outcomes in BRAF-V600-wild type patients was reported, which authors considered lacked statistical significance because of the small number of patients included. The most frequent toxicity of 2-CdA is myelosuppression (neutropenia, lymphopenia) and prolonged immunosuppression, which may predispose to infections, so prolonged antimicrobial prophylaxis may be required [48]. The combination of cladribine/Ara-C has been proved efficient in pediatric patients with either high-risk or risk-organ–positive LCH that was refractory to standard therapy [50].

Among patients with BRAF V600E-mutated LCH, targeted therapy with BRAF inhibitors (vemurafenib, dabrafenib) represents a novel therapeutic approach. These agents may be preferably considered in patients with critical organ involvement or severe symptomatic disease where a quick response is needed. Their use is limited by toxicities, including a higher risk for primary malignancies [51].

As noted above, histiocytic neoplasms are collectively characterized by dependence on MAPK pathway signaling and, consequently, responsiveness to MEK1/2 inhibition. Cobimetinib, a selective inhibitor of MEK1/2, had marked and durable activity in adults with histiocytic neoplasms regardless of the tumor genotype [52], implying that cobimetinib could be used in the entire spectrum of patients with histiocytosis, including those without BRAF V600 mutations. In the same setting trametinib is also an option [53,54]. Finally, targeted therapy (e.g., crizotinib, pexidartinib, larotrectinib, entrectinib, sirolimus, selpercatinib) could be useful in certain circumstances where other gene mutations have been identified [43].

## 5. Conclusions

The skeleton is a frequently involved organ in LCH and severe disability and discomfort can be induced in a patient harboring such lesions. Although the symptoms and signs can be impressive in some cases a careful, patient-specific approach is suggested to avoid unnecessary extensive surgical interventions and/or medical treatment. Both gene-targeted treatment, as well as therapeutic modalities focusing on the specific cytokine environment of the LCH osseous lesion, seem to be the therapeutic approaches that would probably predominate in the future although further research and prospective control studies are urgently needed.

## Figures and Tables

**Figure 1 jcm-11-00909-f001:**
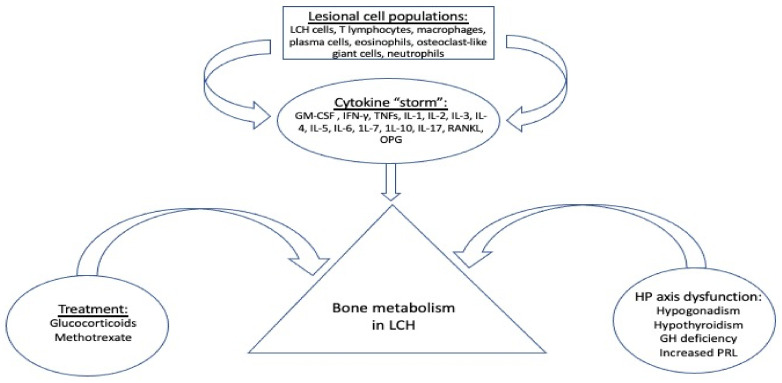
Factors affecting bone metabolism in LCH. GM-CSF: granulocyte-macrophage colony-stimulating factor; OPG: osteoprotegerin; RANKL: receptor activator of nuclear factor kappa-Β ligand; IFN-γ: interferon gamma; TNFs: tumor necrosis factors; IL-: interleukin -; HP: hypothalamus-pituitary; GH: growth hormone; PRL: prolactine.

**Figure 2 jcm-11-00909-f002:**
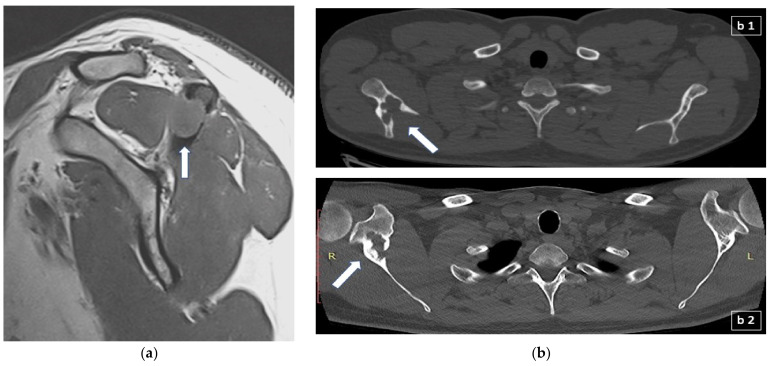
(**a**) Sagittal T1-weighted MR scan showing a 1.5 cm lytic lesion with soft mass tissue extension (arrow) which disrupts the cortex of the left acromion and extents mainly to the supraspinatus muscle; (**b**) appearance of lytic bone lesion in CT: (**b1**) transverse CT scan section showing an extensive lytic lesion of the right scapula (arrow); (**b2**) different transverse CT scan section of the previous lytic lesion of the right scapula, one year later and following treatment with denosumab, depicting now scleroting margins (arrow) suggestive of the healing process. R: right; L: left.

**Table 1 jcm-11-00909-t001:** Therapeutic approach of osseous LCH involvement.

Involvement	Treatment Options
Unifocal bone lesions	Surgical curettage Watchful waiting ^1^ Intralesional injection of corticosteroids Radiation therapy Bisphosphonates
Single system, multiple bone lesions	Bisphosphonates Indomethacin Radiation ^2^ Systemic treatment ^3^
Multisystem disease	Methotrexate Hydroxyurea Cladribine Cytarabine Vinblastine + prednisone BRAF inhibitors MEK inhibitors Other gene-targeted therapy ^4^

^1^ Bone lesions may further resolve following curettage or even simple biopsy which seem to trigger a healing process despite lack of additional treatment. Re-assessment is advised three months after initial evaluation and/or biopsy and every six months afterwards, at least during the next two years. ^2^ Radiation therapy may be used if 1–2 lesions and no response to bisphosphonates. ^3^ Systemic therapy, as in multisystem disease further below, may follow if >2 lesions and no response to bisphosphonates. ^4^ In case of identification of other somatic gene mutations. BRAF: v-raf murine sarcoma viral oncogene homolog B1; MEK: Mitogen-activated Extracellular signal-regulated Kinase.

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
