# Peer review of "Adult Langerhans Cell Histiocytosis and the Skeleton"

_jcm, 2022, doi:10.3390/jcm11040909_

Round 1
Reviewer 1 Report
The authors described bone metabolism, manifestations, evaluation, and treatment in a bone disease of LCH. The authors are not pediatricians, and the treatment strategy presented here was based on “Clinical Practice Guidelines in Oncology: Histiocytic Neoplasms” by NCCN, which describes the treatment of “adult” LCH. However, the authors also described treatment with clofarabine and cladribine / Ara-C used only in children, which is confusing to readers. Therefore, the descriptions should be deleted except for the treatment for adult LCH, and it should be clearly stated that the treatment strategy was for adult LCH according to the above Guidelines. The title should also be changed to “Adult LCH and the skeleton”.
The description of “Bone metabolism in LCH” is easier to understand by adding a figure showing the adverse effects of treatment, hormone deficiency, and osteoclast-activating cytokines on bone metabolism.
Author Response
The authors described bone metabolism, manifestations, evaluation, and treatment in a bone disease of LCH. The authors are not pediatricians, and the treatment strategy presented here was based on “Clinical Practice Guidelines in Oncology: Histiocytic Neoplasms” by NCCN, which describes the treatment of “adult” LCH. However, the authors also described treatment with clofarabine and cladribine / Ara-C used only in children, which is confusing to readers. Therefore, the descriptions should be deleted except for the treatment for adult LCH, and it should be clearly stated that the treatment strategy was for adult LCH according to the above Guidelines. The title should also be changed to “Adult LCH and the skeleton”.
Response: We thank the reviewer for this comment. It is true that all authors are adult physicians. We agree that clofarabine is not used in adults; however, both Ara-C and cladribine has been used among adults and are also suggested in Ref. 43 (now updated with the published paper). We have also changed the title as suggested.
The description of “Bone metabolism in LCH” is easier to understand by adding a figure showing the adverse effects of treatment, hormone deficiency, and osteoclast-activating cytokines on bone metabolism.
Response: A new Figure (now Figure 1) was added as suggested.
Reviewer 2 Report
There are several points of view notes in this manuscript.
1. The abstract emphasizes the background and assessment to show LCH and the presence of a single change to multisystem disease. What is essential to be described by the author(s) is the mechanism of LCH change and the possible therapeutic strategies that can be applied to overcome the problems of LCH.
2. This manuscript also does not strongly lead to the study's objectives to obtain appropriate LCH therapy strategies and good outcomes. For this reason, it is indispensable to affirm the goals and impacts expected in this review.
3. Various inflammatory factors are listed in the section on bone metabolism in LCH lines 87-95. It is well known that some of these factors are inflammatory, and some are anti-inflammatory. How to distinguish the role between these two traits in LCH? Furthermore, it is suggested that what is written in the review are the factors related to and influence the pathophysiology of the disease and the direction of the therapeutic approach of LCH.
4. Author(s) have not done sufficient exploration regarding RANKL's role in the pathogenesis of LCH even though it has been written on lines 104-112. Of course, it must be the author's point of view so that the signaling pathway and approach to LCH therapy can be explained in a straight line.
5. In Table 1, there is a therapeutic approach of osseous LCH involvement. It should be a concern that the treatment option is based on an agreement from a professional organization or as a therapy treatment in a hospital? This information must be clear-cut and accountable. Also, have signaling inhibitors such as BRAF inhibitors and MEK inhibitors been applied to clinical use in a hospital?
6. Optional therapy in table 1 regarding "watchful waiting" and "systemic treatment" is not very clear. Author(s) should detail it carefully.
7. Author(s) must explain carefully the strategy for LCH therapy using pharmacological agents such as bisphosphonates and antiproliferative agents (methotrexate, hydroxyurea, clofarabine, cladribine, etc.), both suspected mechanistic and predictive outcomes.
Author Response
There are several points of view notes in this manuscript.
1. The abstract emphasizes the background and assessment to show LCH and the presence of a single change to multisystem disease. What is essential to be described by the author(s) is the mechanism of LCH change and the possible therapeutic strategies that can be applied to overcome the problems of LCH.
Response: We thank the reviewer for evaluating our manuscript and for all the comments raised further below. We are not sure that we understood the specific comment regarding our Abstract. More specifically, we cannot follow the comments regarding the “LCH change” and therefore we are not able to make any alterations of the Abstract. What we tried to include in the Abstract of this article is a brief description of this review according to the flow of the manuscript (description of the disease and osseous involvement, factors affecting bone metabolism, presentation and evaluation, and finally treatment). We really hope that the reviewer will accept the current form of the Abstract.
- This manuscript also does not strongly lead to the study's objectives to obtain appropriate LCH therapy strategies and good outcomes. For this reason, it is indispensable to affirm the goals and impacts expected in this review.
Response: We thank the reviewer for this comment. We are now emphasizing the objectives of this review at the end of Introduction.
- Various inflammatory factors are listed in the section on bone metabolism in LCH lines 87-95. It is well known that some of these factors are inflammatory, and some are anti-inflammatory. How to distinguish the role between these two traits in LCH? Furthermore, it is suggested that what is written in the review are the factors related to and influence the pathophysiology of the disease and the direction of the therapeutic approach of LCH.
Response: We acknowledge the reviewer’s comment. However, as we are describing a specific component of the disease, namely bone lesions, we think that it is important to list all the factors that have been previously described in the literature within the setting of the cytokine milieu of LCH. Unfortunately, to our knowledge, the role of these cytokines has not been fully elucidated and therefore we are not offering further information regarding the specific impact of each cytokine. In addition, we are not able to offer any specific direction (according to cytokines) for the therapeutic approach of bone lesions in LCH.
- Author(s) have not done sufficient exploration regarding RANKL's role in the pathogenesis of LCH even though it has been written on lines 104-112. Of course, it must be the author's point of view so that the signaling pathway and approach to LCH therapy can be explained in a straight line.
Response: We really thank the reviewer for this comment as our team has extensively discussed the role of RANKL-OPG in the pathogenesis of the disease, while additionally offered and currently tests a new therapeutic approach with an anti-RANKL agent, namely denosumab (check relevant publications: PMID: 22278426, PMID: 25375981, PMID: 28285639).
We are describing the role of RANKL and the possible related therapeutic strategies in lines 99-107 and 197-200.
- In Table 1, there is a therapeutic approach of osseous LCH involvement. It should be a concern that the treatment option is based on an agreement from a professional organization or as a therapy treatment in a hospital? This information must be clear-cut and accountable. Also, have signaling inhibitors such as BRAF inhibitors and MEK inhibitors been applied to clinical use in a hospital?
Response: The therapeutic approaches described in Table 1 are derived from the latest guidelines/recommendations for the disease (Refs 8,43) and can be applied in any health care setting. However, each national health care system has its own rules and legislation for a novel treatment to be implemented but this falls out of the scope of the current review.
- Optional therapy in table 1 regarding "watchful waiting" and "systemic treatment" is not very clear. Author(s) should detail it carefully.
Response: We thank the reviewer for the careful reading and this important notice. We are now clarifying within the footnotes what we mean with "watchful waiting" and "systemic treatment".
- Author(s) must explain carefully the strategy for LCH therapy using pharmacological agents such as bisphosphonates and antiproliferative agents (methotrexate, hydroxyurea, clofarabine, cladribine, etc.), both suspected mechanistic and predictive outcomes.
Response: We are now indicating that agents such as methotrexate, hydroxyurea, clofarabine, cladribine, etc. can be used in LCH as it shares both inflammatory and neoplastic features (lines 205-206). The rationale for the use of bisphosphonates is provided within lines 190-195. Unfortunately, we are not aware of any predictive factors regarding a definitive outcome of its approach, and this is probably the reason for all these different strategies used.
Reviewer 3 Report
The present paper offers a focused updating about the bone involvement in Langerhans cell histiocytosis.
The issue covered is limited (only a specific aspect of a rare disease), but useful for a general view of the topic.
Chapters #1 and specially #2 are a bit too long, and the physiopathologic considerations should be shortened.
Author Response
The present paper offers a focused updating about the bone involvement in Langerhans cell histiocytosis.
The issue covered is limited (only a specific aspect of a rare disease), but useful for a general view of the topic.
Chapters #1 and specially #2 are a bit too long, and the physiopathologic considerations should be shortened.
Response: We appreciate the reviewer’s positive evaluation. We have now reduced chapters 1 and 2 by approximately 220 words.
Reviewer 4 Report
In the paper „Langerhans Cell Histiocytosis and the skeleton“ by Georgakopoulou D., Anastasilakis AD. and Makras P., a comprehensive review, of the research conducted on the topic of Langerhans cell histiocytosis disease effect ond skeleton, is presented. The authors have covered the ethiology and pathology of the disease, and have addressed different aspects of the bone tissue that is being affected. Thus, the review provides quality information and fundamatal knowledge about the influence of Langerhans cell histiocytosis on bone tissue. The authors used appropriate references to support the review.
Small changes to the English language writing should be adopted (commas should be used more frequently).
In the line 31 please reference the correct paper for the orphan disease and cell markers stated in the text.
In the line 52 a spacing should be introduced in front of reference 5.
In the line 238 word “overtreatment” is redundant as you have already used word “extensive” in this same sentence. “Treatment” would be better suited and in line with English writing style.
Author Response
In the paper, Langerhans Cell Histiocytosis and the skeleton“ by Georgakopoulou D., Anastasilakis AD. and Makras P., a comprehensive review, of the research conducted on the topic of Langerhans cell histiocytosis disease effect on skeleton, is presented. The authors have covered the etiology and pathology of the disease, and have addressed different aspects of the bone tissue that is being affected. Thus, the review provides quality information and fundamatal knowledge about the influence of Langerhans cell histiocytosis on bone tissue. The authors used appropriate references to support the review.
Small changes to the English language writing should be adopted (commas should be used more frequently).
In the line 31 please reference the correct paper for the orphan disease and cell markers stated in the text.
In the line 52 a spacing should be introduced in front of reference 5.
In the line 238 word “overtreatment” is redundant as you have already used word “extensive” in this same sentence. “Treatment” would be better suited and in line with English writing style.
Response: We thank the reviewer for his/her positive evaluation. All suggested changes were made accordingly.
Round 2
Reviewer 1 Report
Most have been revised well, but a little more needs to be revised.
#1. It is better to delete “clofarabine” on line 205.
#2. Regarding the description of "The combination of cladribine / Ara-C has been proved efficient in patients with either high-risk or risk-organ–positive LCH that was refractory to standard therapy [50]." on line 216-7, it is better to delete the description. Alternatively, the description should be limited to “in pediatric patients”.
Author Response
We are grateful for the final comments raised by the reviewer. All suggestions have been taken into consideration and necessary changes have been made accordingly and are now highlighted within the text.
Reviewer 2 Report
The revised version of the manuscript entitled "Adult Langerhans Cell Histiocytosis and Skeleton" has undergone significant improvements. There are also supported by replacing some more precise points to clarify the sentence. In addition, the author(s) has clarified some issues very well. Finally, they have also answered some of the questions or comments discussed in previous reviews, although not all of them.
Furthermore, I recommend accepting this revised version of the manuscript.
Author Response
We appreciate the positive comments of the reviewer and we are grateful for the previous helpful suggestions.